# HIGHER ORDER RECURRENT NEURAL NETWORKS

**Rohollah Soltani & Hui Jiang**
Department of Computer Science and Engineering
York University
Toronto, CA
`{rsoltani,hj}@cse.yorku.ca`

## ABSTRACT

In this paper, we study novel neural network structures to better model long term dependency in sequential data. We propose to use more memory units to keep track of more preceding states in recurrent neural networks (RNNs), which are all recurrently fed to the hidden layers as feedback through different weighted paths. By extending the popular recurrent structure in RNNs, we provide the models with better short-term memory mechanism to learn long term dependency in sequences. Analogous to digital filters in signal processing, we call these structures as higher order RNNs (HORNNs). Similar to RNNs, HORNNs can also be learned using the back-propagation through time method. HORNNs are generally applicable to a variety of sequence modelling tasks. In this work, we have examined HORNNs for the language modeling task using two popular data sets, namely the Penn Treebank (PTB) and English text8. Experimental results have shown that the proposed HORNNs yield the state-of-the-art performance on both data sets, significantly outperforming the regular RNNs as well as the popular LSTMs.

## 1 INTRODUCTION

In the recent resurgence of neural networks in deep learning, deep neural networks have achieved successes in various real-world applications, such as speech recognition, computer vision and natural language processing. Deep neural networks (DNNs) with a deep architecture of multiple nonlinear layers are an expressive model that can learn complex features and patterns in data. Each layer of DNNs learns a representation and transfers them to the next layer and the next layer may continue to extract more complicated features, and finally the last layer generates the desirable output. From early theoretical work, it is well known that neural networks may be used as the universal approximators to map from any fixed-size input to another fixed-size output. Recently, more and more empirical results have demonstrated that the deep structure in DNNs is not just powerful in theory but also can be reliably learned in practice from a large amount of training data.

Sequential modeling is a challenging problem in machine learning, which has been extensively studied in the past. Recently, many deep neural network based models have been successful in this area, as shown in various tasks such as language modeling Mikolov (2012), sequence generation Graves (2013); Sutskever et al. (2011), machine translation Sutskever et al. (2014) and speech recognition Graves et al. (2013). Among various neural network models, recurrent neural networks (RNNs) are appealing for modeling sequential data because they can capture long term dependency in sequential data using a simple mechanism of recurrent feedback. RNNs can learn to model sequential data over an extended period of time, then carry out rather complicated transformations on the sequential data. RNNs have been theoretically proved to be a turing complete machine Siegelmann & Sontag (1995). RNNs in principle can learn to map from one variable-length sequence to another. When unfolded in time, RNNs are equivalent to very deep neural networks that share model parameters and receive the input at each time step. The recursion in the hidden layer of RNNs can act as an excellent memory mechanism for the networks. In each time step, the learned recursion weights may decide what information to discard and what information to keep in order to relay onwards along time. While RNNs are theoretically powerful, the learning of RNNs needs to use the back-propagation through time (BPTT) method Werbos (1990) due to the internal recurrent cycles. Unfortunately, in practice, it turns out to be rather difficult to train RNNs to capture long-term dependency due to the fact that

the gradients in BPTT tend to either vanish or explode Bengio et al. (1994). Many heuristic methods have been proposed to solve these problems. For example, a simple method, called *gradient clipping*, is used to avoid gradient explosion Mikolov (2012). However, RNNs still suffer from the vanishing gradient problem since the gradients decay gradually as they are back-propagated through time. As a result, some new recurrent structures are proposed, such as long short-term memory (LSTM) Hochreiter & Schmidhuber (1997) and gated recurrent unit (GRU) Cho et al. (2014). These models use some learnable gates to implement rather complicated feedback structures, which ensure that some feedback paths can allow the gradients to flow back in time effectively. These models have given promising results in many practical applications, such as sequence modeling Graves (2013), language modeling Sundermeyer et al. (2012), hand-written character recognition Liwicki et al. (2012), machine translation Cho et al. (2014), speech recognition Graves et al. (2013).

In this paper, we explore an alternative method to learn recurrent neural networks (RNNs) to model long term dependency in sequential data. We propose to use more memory units to keep track of more preceding RNN states, which are all recurrently fed to the hidden layers as feedback through different weighted paths. Analogous to digital filters in signal processing, we call these new recurrent structures as higher order recurrent neural networks (HORNNs). At each time step, the proposed HORNNs directly combine multiple preceding hidden states from various history time steps, weighted by different matrices, to generate the feedback signal to each hidden layer. By aggregating more history information of the RNN states, HORNNs are provided with better short-term memory mechanism than the regular RNNs. Moreover, those direct connections to more previous RNN states allow the gradients to flow back smoothly in the BPTT learning stage. All of these ensure that HORNNs can be more effectively learned to capture long term dependency. Similar to RNNs and LSTMs, the proposed HORNNs are general enough for variety of sequential modeling tasks. In this work, we have evaluated HORNNs for the language modeling task on two popular data sets, namely the Penn Treebank (PTB) and English text8 sets. Experimental results have shown that HORNNs yield the state-of-the-art performance on both data sets, significantly outperforming the regular RNNs as well as the popular LSTMs.

## 2 RELATED WORK

Hierarchical recurrent neural network proposed in Hihi & Bengio (1996) is one of the earliest papers that attempt to improve RNNs to capture long term dependency in a better way. It proposes to add linear time delayed connections to RNNs to improve the gradient descent learning algorithm to find a better solution, eventually solving the gradient vanishing problem. However, in this early work, the idea of multi-resolution recurrent architectures has only been preliminarily examined for some simple small-scale tasks. This work is somehow relevant to our work in this paper but the higher order RNNs proposed here differs in several aspects. Firstly, we propose to use weighted connections in the structure, instead of simple multi-resolution short-cut paths. This makes our models fall into the category of higher order models. Secondly, we have proposed to use various pooling functions in generating the feedback signals, which is critical in normalizing the dynamic ranges of gradients flowing from various paths. Our experiments have shown that the success of our models is largely attributed to this technique.

The most successful approach to deal with vanishing gradients so far is to use long short term memory (LSTM) model Hochreiter & Schmidhuber (1997). LSTM relies on a fairly sophisticated structure made of gates to control flow of information to the hidden neurons. The drawback of the LSTM is that it is complicated and slow to learn. The complexity of this model makes the learning very time consuming, and hard to scale for larger tasks. Another approach to address this issue is to add a hidden layer to RNNs Mikolov et al. (2014). This layer is responsible for capturing longer term dependencies in input data by making its weight matrix close to identity. Recently, clockwork RNNs Koutnik et al. (2014) are proposed to address this problem as well, which splits each hidden layer into several modules running at different clocks. Each module receives signals from input and computes its output at a predefined clock rate. Gated feedback recurrent neural networks Chung et al. (2015) attempt to implement a generalized version using the gated feedback connection between layers of stacked RNNs, allowing the model to adaptively adjust the connection between consecutive hidden layers.

Besides, short-cut skipping connections were considered earlier in Wermter (1992), and more recently have been found useful in learning very deep feed-forward neural networks as well, such as Lee et al. (2014); He et al. (2015). These skipping connections between various layers of neural networks can improve the flow of information in both forward and backward passes. Among them, highway networks Srivastava et al. (2015) introduce rather sophisticated skipping connections between layers, controlled by some gated functions.

## 3 HIGHER ORDER RECURRENT NEURAL NETWORKS

A recurrent neural network (RNN) is a type of neural network suitable for modeling a sequence of arbitrary length. At each time step $t$, an RNN receives an input $\mathbf{x}_t$, the state of the RNN is updated recursively as follows (as shown in the left part of Figure 1):

$$\mathbf{h}_t = f(W_{in}\mathbf{x}_t + W_h\mathbf{h}_{t-1}) \tag{1}$$

where $f(\cdot)$ is an nonlinear activation function, such as sigmoid or rectified linear (ReLU), and $W_{in}$ is the weight matrix in the input layer and $W_h$ is the state to state recurrent weight matrix. Due to the recursion, this hidden layer may act as a short-term memory of all previous input data.

Given the state of the RNN, i.e., the current activation signals in the hidden layer $\mathbf{h}_t$, the RNN generates the output according to the following equation:

$$\mathbf{y}_t = g(W_{out}\mathbf{h}_t) \tag{2}$$

where $g(\cdot)$ denotes the softmax function and $W_{out}$ is the weight matrix in the output layer. In principle, this model can be trained using the back-propagation through time (BPTT) algorithm Werbos (1990). This model has been used widely in sequence modeling tasks like language modeling Mikolov (2012).

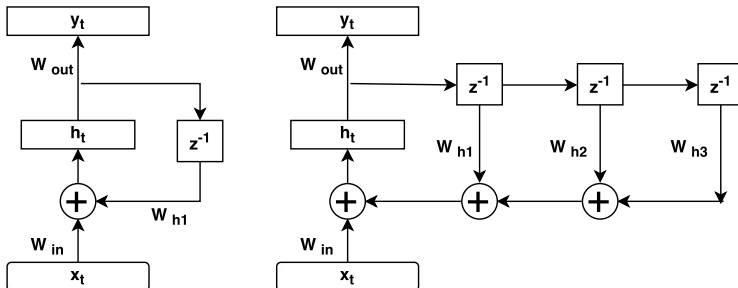

Figure 1: Comparison of model structures between an RNN (1st order) and a higher order RNN (3rd order). The symbol $z^{-1}$ denotes a time-delay unit (equivalent to a memory unit).

### 3.1 HIGHER ORDER RNNS (HORNNS)

RNNs are very deep in time and the hidden layer at each time step represents the entire input history, which acts as a short-term memory mechanism. However, due to the gradient vanishing problem in back-propagation, it turns out to be very difficult to learn RNNs to model long-term dependency in sequential data.

In this paper, we extend the standard RNN structure to better model long-term dependency in sequential data. As shown in the right part of Figure 1, instead of using only the previous RNN state as the feedback signal, we propose to employ multiple memory units to generate the feedback signal at each time step by directly combining multiple preceding RNN states in the past, where these time-delayed RNN states go through separate feedback paths with different weight matrices. Analogous to the filter structures used in signal processing, we call this new recurrent structure as *higher order RNNs*, HORNNs in short. The order of HORNNs depends on the number of memory units used for feedback. For example, the model used in the right of Figure 1 is a 3rd-order HORNN. On the other hand, regular RNNs may be viewed as 1st-order HORNNs.

In HORNNs, the feedback signal is generated by combining multiple preceding RNN states. There-fore, the state of an $N$-th order HORNN is recursively updated as follows:

$$\mathbf{h}_t = f\left(W_{in}\mathbf{x}_t + \sum_{n=1}^{N} W_{hn}\mathbf{h}_{t-n}\right) \tag{3}$$

where $\{W_{hn} \mid n = 1, \cdots N\}$ denotes the weight matrices used for various feedback paths. Similar to

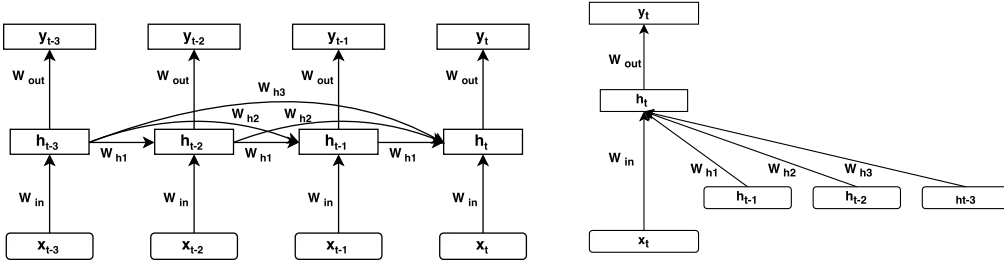
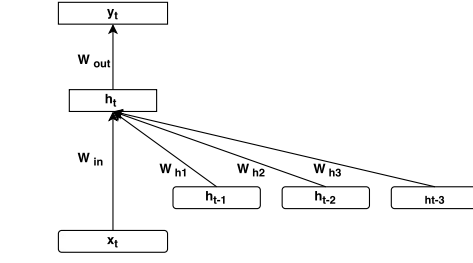

Figure 2: Unfolding a 3rd-order HORNN

Figure 3: Illustration of all back-propagation paths in BPTT for a 3rd-order HORNN.

RNNs, HORNNs can also be unfolded in time to get rid of the recurrent cycles. As shown in Figure 2, we unfold a 3rd-order HORNN in time, which clearly shows that each HORNN state is explicitly decided by the current input $\mathbf{x}_t$ and all previous 3 states in the past. This structure looks similar to the skipping short-cut paths in deep neural networks but each path in HORNNs maintains a learnable weight matrix. The new structure in HORNNs can significantly improve the model capacity to cap-ture long-term dependency in sequential data. At each time step, by explicitly aggregating multiple preceding hidden activities, HORNNs may derive a good representation of the history information in sequences, leading to a significantly enhanced short-term memory mechanism.

During the backprop learning procedure, these skipping paths directly connected to more previous hidden states of HORNNs may allow the gradients to flow more easily back in time, which even-tually leads to a more effective learning of models to capture long term dependency in sequences. Therefore, this structure may help to largely alleviate the notorious problem of vanishing gradients in the RNN learning.

Obviously, HORNNs can be learned using the same BPTT algorithm as regular RNNs, except that the error signals at each time step need to be back-propagated to multiple feedback paths in the network. As shown in Figure 3, for a 3rd-order HORNN, at each time step $t$, the error signal from the hidden layer $\mathbf{h}_t$ will have to be back-propagated into four different paths: i) the first one back to the input layer, $\mathbf{x}_t$; ii) three more feedback paths leading to three different histories in time scales, namely $\mathbf{h}_{t-1}$, $\mathbf{h}_{t-2}$ and $\mathbf{h}_{t-3}$.

Interestingly enough, if we use a fully-unfolded implementation for HORNNs as in Figure 2, the overall computation complexity is comparable with regular RNNs. Given a whole sequence, we may first simultaneously compute all hidden activities (from $\mathbf{x}_t$ to $\mathbf{h}_t$ for all $t$). Secondly, we recursively update $\mathbf{h}_t$ for all $t$ using eq.(3). Finally, we use GPUs to compute all outputs together from the updated hidden states (from $\mathbf{h}_t$ to $\mathbf{y}_t$ for all $t$) based on eq.(2). The backward pass in learning can also be implemented in the same three-step procedure. Except the recursive updates in the second step (this issue remains the same in regular RNNs), all remaining computation steps can be formulated as large matrix multiplications. As a result, the computation of HORNNs can be implemented fairly efficiently using GPUs.

## 3.2 POOLING FUNCTIONS FOR HORNNs

As discussed above, the shortcut paths in HORNNs may help the models to capture long-term de-pendency in sequential data. On the other hand, they may also complicate the learning in a different way. Due to different numbers of hidden layers along various paths, the signals flowing from differ-ent paths may vary dramatically in the dynamic range. For example, in the forward pass in Figure 2, three different feedback signals from different time scales, e.g. $\mathbf{h}_{t-1}$, $\mathbf{h}_{t-2}$ and $\mathbf{h}_{t-3}$, flow into

the hidden layer to compute the new hidden state $\mathbf{h}_t$. The dynamic range of these signals may vary dramatically from case to case. The situation may get even worse in the backward pass during the BPTT learning. For example, in a 3rd-order HORNN in Figure 2, the node $\mathbf{h}_{t-3}$ may directly receive an error signal from the node $\mathbf{h}_t$. In some cases, it may get so strong as to overshadow other error signals coming from closer neighbours of $\mathbf{h}_{t-1}$ and $\mathbf{h}_{t-2}$. This may impede the learning of HORNNs, yielding slow convergence or even poor performance.

Here, we have proposed to use some pooling functions to calibrate the signals from different feedback paths before they are used to recursively generate a new hidden state, as shown in Figure 4. In the following, we will investigate three different choices for the pooling function in Figure 4, including *max*-based pooling, FOFE-based pooling and gated pooling.

### 3.2.1 MAX-BASED POOLING

Max-based pooling is a simple strategy that chooses the most responsive unit (exhibiting the largest activation value) among various paths to transfer to the hidden layer to generate the new hidden state. Many biological experiments have shown that biological neuron networks tend to use a similar strategy in learning and firing.

In this case, instead of using eq.(3), we use the following formula to update the hidden state of HORNNs:

$$\mathbf{h}_t = f\left(W_{in}\mathbf{x}_t + \max_{n=1}^{N}\left(W_{hn}\mathbf{h}_{t-n}\right)\right) \tag{4}$$

where maximization is performed element-wisely to choose the maximum value in each dimension to feed to the hidden layer to generate the new hidden state. The aim here is to capture the most relevant feature and map it to a fixed predefined size.

The max pooling function is simple and biologically inspired. However, the max pooling strategy also has some serious disadvantages. For example, it has no forgetting mechanism and the signals may get stronger and stronger. Furthermore, it loses the order information of the preceding histories since it only choose the maximum values but it does not know where the maximum comes from.

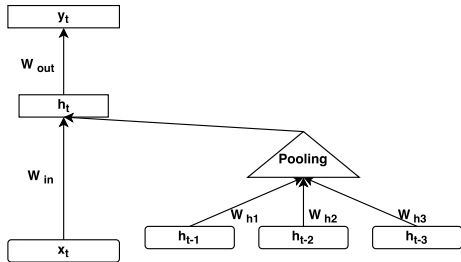
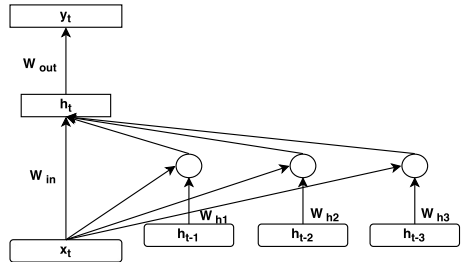

Figure 4: A pooling function is used to calibrate various feedback paths in HORNNs.

Figure 5: Gated HORNNs use learnable gates to combine various feedback signals.

### 3.2.2 FOFE-BASED POOLING

The fixed-size ordinally-forgetting encoding (FOFE) method was proposed in Zhang et al. (2015) to encode any variable-length sequence of data into a fixed-size representation. In FOFE, a single forgetting factor $\alpha$ ($0 < \alpha < 1$) is used to encode the position information in sequences based on the idea of exponential forgetting to derive invertible fixed-size representations. In this work, we borrow this simple idea of exponential forgetting to calibrate all preceding histories using a pre-selected forgetting factor as follows:

$$\mathbf{h}_t = f\left(W_{in}\mathbf{x}_t + \sum_{n=1}^{N} \alpha^n \cdot W_{hn}\mathbf{h}_{t-n}\right) \tag{5}$$

where the forgetting factor $\alpha$ is manually pre-selected between $0 < \alpha < 1$. The above constant coefficients related to $\alpha$ play an important role in calibrating signals from different paths in both

forward and backward passes of HORNNs since they slightly underweight the older history over the recent one in an explicit way.

### 3.2.3 GATED HORNNS

In this section, we follow the ideas of the learnable gates in LSTMs Hochreiter & Schmidhuber (1997) and GRUs Cho et al. (2014) as well as the recent soft-attention in Bahdanau et al. (2014). Instead of using constant coefficients derived from a forgetting factor, we may let the network automatically determine the combination weights based on the current state and input. In this case, we may use sigmoid gates to compute combination weights to regulate the information flowing from various feedback paths. The sigmoid gates take the current data and previous hidden state as input to decide how to weight all of the precede hidden states. The gate function weights how the current hidden state is generated based on all the previous time-steps of the hidden layer. This allows the network to potentially remember information for a longer period of time. In a gated HORNN, the hidden state is recursively computed as follows:

$$\mathbf{h}_t = f \left( W_{in}\mathbf{x}_t + \sum_{n=1}^{N} \mathbf{r}_n \odot \left( W_{hn}\mathbf{h}_{t-n} \right) \right) \tag{6}$$

where $\odot$ denotes element-wise multiplication of two equally-sized vectors, and the gate signal $\mathbf{r}_n$ is calculated as

$$\mathbf{r}_n = \sigma \left( W_{1n}^g \mathbf{x}_t + W_{2n}^g \mathbf{h}_{t-n} \right) \tag{7}$$

where $\sigma(\cdot)$ is the sigmoid function, and $W_{1n}^g$ and $W_{2n}^g$ denote two weight matrices introduced for each gate.

Note that the computation complexity of gated HORNNs is comparable with LSTMs and GRUs, significantly exceeding the other HORNN structures because of the overhead from the gate functions in eq. (7).

## 4 EXPERIMENTS

In this section, we evaluate the proposed higher order RNNs (HORNNs) on several language modeling tasks. A statistical language model (LM) is a probability distribution over sequences of words in natural languages. Recently, neural networks have been successfully applied to language modeling Bengio et al. (2003); Mikolov et al. (2011), yielding the state-of-the-art performance. In language modeling tasks, it is quite important to take advantage of the long-term dependency of natural languages. Therefore, it is widely reported that RNN based LMs can outperform feedforward neural networks in language modeling tasks. We have chosen two popular LM data sets, namely the Penn Treebank (PTB) and English text8 sets, to compare our proposed HORNNs with traditional n-gram LMs, RNN-based LMs and the state-of-the-art performance obtained by LSTMs Graves (2013); Mikolov et al. (2014), FOFE based feedforward NNs Zhang et al. (2015) and memory networks Sukhbaatar et al. (2015).

In our experiments, we use the mini-batch stochastic gradient decent (SGD) algorithm to train all neural networks. The number of back-propagation through time (BPTT) steps is set to 30 for all recurrent models. Each model update is conducted using a mini-batch of 20 subsequences, each of which is of 30 in length. All model parameters (weight matrices in all layers) are randomly initialized based on a Gaussian distribution with zero mean and standard deviation of 0.1. A hard clipping is set to 5.0 to avoid gradient explosion during the BPTT learning. The initial learning rate is set to 0.5 and we halve the learning rate at the end of each epoch if the cross entropy function on the validation set does not decrease. We have used the weight decay, momentum and column normalization Pachitariu & Sahani (2013) in our experiments to improve model generalization. In the FOFE-based pooling function for HORNNs, we set the forgetting factor, $\alpha$, to 0.6. We have used 400 nodes in each hidden layer for the PTB data set and 500 nodes per hidden layer for the English text8 set. In our experiments, we do not use the dropout regularization Zaremba et al. (2014) in all experiments since it significantly slows down the training speed, not applicable to any larger corpora. [1]

---

[1] We will soon release the code for readers to reproduce all results reported in this paper.

Table 1: Perplexities on the PTB test set for various HORNNs are shown as a function of order (2, 3, 4). Note the perplexity of a regular RNN (1st order) is 123, as reported in Mikolov et al. (2011).

| Models | $2^{nd}$ order | $3^{rd}$ order | $4^{th}$ order |
|---|---|---|---|
| HORNN | 111 | 108 | 109 |
| Max HORNN | 110 | 109 | 108 |
| FOFE HORNN | 103 | 101 | 100 |
| Gated HORNN | 102 | 100 | 100 |

## 4.1 LANGUAGE MODELING ON PTB

The standard Penn Treebank (PTB) corpus consists of about 1M words. The vocabulary size is limited to 10k. The preprocessing method and the way to split data into training/validation/test sets are the same as Mikolov et al. (2011). PTB is a relatively small text corpus. We first investigate various model configurations for the HORNNs based on PTB and then compare the best performance with other results reported on this task.

### 4.1.1 EFFECT OF ORDERS IN HORNNS

In the first experiment, we first investigate how the used orders in HORNNs may affect the performance of language models (as measured by perplexity). We have examined all different higher order model structures proposed in this paper, including HORNNs and various pooling functions in HORNNs. The orders of these examined models varies among 2, 3 and 4. We have listed the performance of different models on PTB in Table 1. As we may see, we are able to achieve a significant improvement in perplexity when using higher order RNNs for language models on PTB, roughly 10-20 reduction in PPL over regular RNNs. We can see that performance may improve slightly when the order is increased from 2 to 3 but no significant gain is observed when the order is further increased to 4. As a result, we choose the 3rd-order HORNN structure for the following experiments. Among all different HORNN structures, we can see that FOFE-based pooling and gated structures yield the best performance on PTB.

In language modeling, both input and output layers account for the major portion of model parameters. Therefore, we do not significantly increase model size when we go to higher order structures. For example, in Table 1, a regular RNN contains about 8.3 millions of weights while a 3rd-order HORNN (the same for max or FOFE pooling structures) has about 8.6 millions of weights. In comparison, an LSTM model has about 9.3 millions of weights and a 3rd-order gated HORNN has about 9.6 millions of weights.

As for the training speed, most HORNN models are only slightly slower than regular RNNs. For example, one epoch of training on PTB running in one NVIDIA's TITAN X GPU takes about 80 seconds for an RNN, about 120 seconds for a 3rd-order HORNN (the same for max or FOFE pooling structures). Similarly, training of gated HORNNs is also slightly slower than LSTMs. For example, one epoch on PTB takes about 200 seconds for an LSTM, and about 225 seconds for a 3rd-order gates HORNN.

### 4.1.2 MODEL COMPARISON ON PENN TREEBANK

At last, we report the best performance of various HORNNs on the PTB test set in Table 2. We compare our 3rd-order HORNNs with all other models reported on this task, including RNN Mikolov et al. (2011), stack RNN Pascanu et al. (2014), deep RNN Pascanu et al. (2014), FOFE-FNN Zhang et al. (2015) and LSTM Graves (2013). [2] From the results in Table 2, we can see that our proposed higher order RNN architectures significantly outperform all other baseline models reported on this task. Both FOFE-based pooling and gated HORNNs have achieved the state-of-the-art performance,

---

[2] All models in Table 2 do not use the dropout regularization, which is somehow equivalent to data augmentation. In Zaremba et al. (2014); Kim et al. (2015), the proposed LSTM-LMs (word level or character level) achieve lower perplexity but they both use the dropout regularization and much bigger models and it takes days to train the models, which is not applicable to other larger tasks.

Table 2: Perplexities on the PTB test set for various examined models.

| Models | Test |
| --- | --- |
| KN 5-gram Mikolov et al. (2011) | 141 |
| RNN Mikolov et al. (2011) | 123 |
| CSLM5Aransa et al. (2015) | 118.08 |
| LSTM Graves (2013) | 117 |
| genCNN Wang et al. (2015) | 116.4 |
| Gated word&charMiyamoto & Cho (2016) | 113.52 |
| E2E Mem Net Sukhbaatar et al. (2015) | 111 |
| Stack RNN Pascanu et al. (2014) | 110 |
| Deep RNN Pascanu et al. (2014) | 107 |
| FOFE-FNN Zhang et al. (2015) | 108 |
| HORNN ($3^{rd}$ order) | 108 |
| Max HORNN ($3^{rd}$ order) | 109 |
| FOFE HORNN ($3^{rd}$ order) | **101** |
| Gated HORNN ($3^{rd}$ order) | **100** |

Table 3: Perplexities on the text8 test set for various models.

| Models | Test |
| --- | --- |
| RNN Mikolov et al. (2014) | 184 |
| LSTM Mikolov et al. (2014) | 156 |
| SCRNN Mikolov et al. (2014) | 161 |
| E2E Mem Net Sukhbaatar et al. (2015) | 147 |
| HORNN ($3^{rd}$ order) | 172 |
| Max HORNN ($3^{rd}$ order) | 163 |
| FOFE HORNN ($3^{rd}$ order) | 154 |
| Gated HORNN ($3^{rd}$ order) | **144** |

i.e., 100 in perplexity on this task. To the best of our knowledge, this is the best reported performance on PTB under the same training condition.

## 4.2 LANGUAGE MODELING ON ENGLISH TEXT8

In this experiment, we will evaluate our proposed HORNNs on a much larger text corpus, namely the English text8 data set. The text8 data set contains a preprocessed version of the first 100 million characters downloaded from the Wikipedia website. We have used the same preprocessing method as Mikolov et al. (2014) to process the data set to generate the training and test sets. We have limited the vocabulary size to about 44k by replacing all words occurring less than 10 times in the training set with an <UNK> token. The text8 set is about 20 times larger than PTB in corpus size. The model training on text8 takes longer to finish. We have not tuned hyperparameters in this data set. We simply follow the best setting used in PTB to train all HORNNs for the text8 data set. Meanwhile, we also follow the same learning schedule used in Mikolov et al. (2014): We first initialize the learning rate to 0.5 and run 5 epochs using this learning rate; After that, the learning rate is halved at the end of every epoch.

Because the training is time-consuming, we have only evaluated 3rd-order HORNNs on the text8 data set. The perplexities of various HORNNs are summarized in Table 3. We have compared our HORNNs with all other baseline models reported on this task, including RNN Mikolov et al. (2014), LSTM Mikolov et al. (2014), SCRNN Mikolov et al. (2014) and end-to-end memory networks Sukhbaatar et al. (2015). Results have shown that all HORNN models work pretty well in this data set except the normal HORNN significantly underperforms the other three models. Among them, the gated HORNN model has achieved the best performance, i.e., 144 in perplexity on this task, which is slightly better than the recent result obtained by end-to-end memory networks (using a rather complicated structure). To the best of our knowledge, this is the best performance reported on this task.

## 5 CONCLUSIONS

In this paper, we have proposed some new structures for recurrent neural networks, called as *higher order RNNs (HORNNs)*. In these structures, we use more memory units to keep track of more preceding RNN states, which are all fed along various feedback paths to the hidden layer to generate the feedback signals. In this way, we may enhance the model to capture long term dependency in sequential data. Moreover, we have proposed to use several types of pooling functions to calibrate multiple feedback paths. Experiments have shown that the pooling technique plays a critical role in learning higher order RNNs effectively. In this work, we have examined HORNNs for the language modeling task using two popular data sets, namely the Penn Treebank (PTB) and text8 sets. Experimental results have shown that the proposed higher order RNNs yield the state-of-the-art per-

formance on both data sets, significantly outperforming the regular RNNs as well as the popular LSTMs. As the future work, we are going to continue to explore HORNNs for other sequential modeling tasks, such as speech recognition, sequence-to-sequence modelling and so on.

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
