# Peer review of "Higher Order Recurrent Neural Networks"

_ICLR 2017 — rejected_

[Official Review · AnonReviewer2 · rating 3 · confidence 4 · 17 Dec 2016 (modified: 18 Dec 2016)]
**Incremental work**

This paper proposes an idea of looking n-steps backward when modelling sequences with RNNs. The proposed RNN does not only use the previous hidden state (t-1) but also looks further back ( (t - k) steps, where k=1,2,3,4 ). The paper also proposes a few different ways to aggregate multiple hidden states from the past.


The reviewer can see few issues with this paper.

Firstly, the writing of this paper requires improvement. The introduction and abstract are wasting too much space just to explain unrelated facts or to describe already well-known things in the literature. Some of the statements written in the paper are misleading. For instance, it explains, “Among various neural network models, recurrent neural networks (RNNs) are appealing for modeling sequential data because they can capture long term dependency in sequential data using a simple mechanism of recurrent feedback” and then it says RNNs cannot actually capture long-term dependencies that well. RNNs are appealing in the first place because they can handle variable length sequences and can model temporal relationships between each symbol in a sequence. The criticism against LSTMs is hard to accept when it says: LSTMs are slow and because of the slowness, they are hard to scale at larger tasks. But we all know that some companies are already using gigantic seq2seq models for their production (LSTMs are used as building blocks in their systems). This indicates that the LSTMs can be practically used in a very large-scale setting.


Secondly, the idea proposed in the paper is incremental and not new to the field. There are other previous works that propose to use direct connections to the previous hidden states [1]. However, the previous works do not use aggregation of multiple number of previous hidden states. Most importantly, the paper fails to deliver a proper analysis on whether its main contribution is actually helpful to improve the problem posed in the paper. The new architecture is said that it handles the long-term dependencies better, however, there is no rigorous proof or intuitive design in the architecture that help us to understand why it should work better. By the design of the architecture, and speaking in very high-level, it seems like the model maybe helpful to mitigate the vanishing gradients issue by a linear factor. It is always a good practice to have at least one page to analyze the empirical findings in the paper.


Thirdly, the baseline models used in this paper are very weak. Their are plenty of other models that are trained and tested on word-level language modelling task using Penn Treebank corpus, but the paper only contains a few of outdated models. I cannot fully agree on the statement “To the best of our knowledge, this is the best performance on PTB under the same training condition”, these days, RNN-based methods usually score below 80 in terms of the test perplexity, which are far lower than 100 achieved in this paper.


[1] Zhang et al., “Architectural Complexity Measures of Recurrent Neural Networks”, NIPS’16

[Official Review · AnonReviewer3 · rating 6 · confidence 4 · 18 Dec 2016]
**can be improved**

I think the backbone of the paper is interesting and could lead to something potentially quite useful. I like the idea of connecting signal processing with recurrent network and then using tools from one setting in the other. However, while the work has nuggets of very interesting observations, I feel they can be put together in a better way. 
I think the writeup and everything can be improved and I urge the authors to strive for this if the paper doesn't go through. I think some of the ideas of how to connect to the past are interesting, it would be nice to have more experiments or to try to understand better why this connections help and how.

[Official Review · AnonReviewer1 · rating 4 · confidence 4 · 20 Dec 2016]
**Interesting idea, but not ready yet**

The authors of the paper explore the idea of incorporating skip connections *over time* for RNNs. Even though the basic idea is not particularly innovative, a few proposals on how to merge that information into the current hidden state with different pooling functions are evaluated. The different models are compared on two popular text benchmarks.

Some points.

1) The experiments feature only NLP and only prediction tasks. It would have been nice to see the models in other domains, i.e. modelling a conditional distribution p(y|x), not only p(x). Further, sensory input data such as audio or video would have given further insight.

2) As pointed out by other reviewers, it does not feel as if the comparisons to other models are fair. SOTA on NLP changes quickly and it is hard to place the experiments in the complete picture.

3) It is claimed that this helps long-term prediction. I think the paper lacks a corresponding analysis, as pointed out in an earlier question of mine.

4)  It is claimed that LSTM trains slow and is hard to scale. For one does this not match my personal experience. Then, the prevalence of LSTM systems in production systems (e.g. Google, Baidu, Microsoft, …) clearly speaks against this.


I like the basic idea of the paper, but the points above make me think it is not ready for publication.

[Final Decision · Program Chairs · 06 Feb 2017]
**ICLR committee final decision**

Paper presents the idea of using higher order recurrence in LSTMs. The ideas are well presented and easy to follow.
 However, the results are far from convincing, easily being below well established numbers in the domain. Since the mode is but a very simple extension of the baseline recurrent models using LSTMs that are state of the art on language modelling, it should have been easy to make a fair comparison by replacing the LSTMs of the state of the art models with higher order versions, but the authors did not do that. Their claimed numbers for SOTA are far from previously reported numbers in that setup, as pointed out by reviewers.